# An approach to optimizing dietary protein to growth and body composition in walking catfish, *Clarias batrachus* (Linneaeus, 1758)

Zara Naeem[1], Amina Zuberi[1], Muhammad Ali[2], Ammar Danyal Naeem[3], Muhammad Naeem[3] *

1 Fisheries and Aquaculture Program, Department of Zoology, Faculty of Biological Sciences, Quaid-i-Azam University, Islamabad, Pakistan, 2 Vice Chancellor, Quaid-i-Azam University, Islamabad, Pakistan, 3 Institute of Zoology, Bahauddin Zakariya University, Multan, Pakistan

* dr_naeembzu@yahoo.com

**Data Availability Statement:** All relevant data are within the paper.

**Funding:** The author(s) received no specific funding for this work.

## Abstract

*Clarias batrachus* is a commercially important food fish. In the present study, effect of varying dietary protein levels was evaluated on the survival, growth parameters and proximate composition of *C. batrachus*. Diets comprising 25%, 30%, 35%, 40%, 45%, and 50% crude protein (CP) were supplied to fish in T1, T2, T3, T4, T5, and T6, respectively, at the rate of 5% of fish body weight for the entire 90 days, twice daily. Size of each stocked *C. batrachus* was recorded after 15 days. Results revealed 100% survival rate of *C. batrachus* in all treatments. Significantly highest (P<0.001) mean value of weight gain (g/fish), percent weight gain, daily growth rate, specific growth rate and protein efficiency ratio (PER) in *C. batrachus* were recorded, reared in T4 by feeding 40% CP in diet. The best FCR value (1.90±0.02) for *C. batrachus* was obtained in T4 by feeding 40%CP in diet. Mean value of water, ash, fat and protein contents (wet mass) were ranged 74.10–79.23%, 3.12–4.68%, 3.90–4.43% and 13.09–16.79% for *C. batrachus* in the studied treatment groups. Water content (%) was found significantly (P<0.05) higher in the body of *C. batrachus* for T1, T2, T3 and T6 than for T4 and T5. Ash was found significantly (P<0.05) higher in the fish reared in T4 and T5. Fat content in the wet body mass of *C. batrachus* was found significantly higher in T4 and T1. While, significant higher (P<0.05) values of mean protein content was noted in *C. batrachus* reared in T4 and T5. Body composition of *C. batrachus* was also categorically affected by body size, however, condition factor showed non-significant correlation in most of the relationships in the present study. Overall, results indicated that feeding appropriate diet (containing 40% CP) to the fish resulted good growth performance, lower FCR and higher protein content in the fish. Present study provides valuable knowledge of optimal dietary protein level in *C. batrachus* which will help in commercial success of aquaculture.

## 1. Introduction

Walking catfish, *Clarias batrachus*, is one of the most well-known catfish species and a common food fish due to its lack of intramuscular bones, distinctive flavour and excellent

**Competing interests:** The authors have declared that no competing interests exist.

nutritional content. Moreover, this fish can be easily stored and transported alive to markets. As a result, consumers are always eager to pay more for this fish [1]. This commercially important fish species is widely used as an aquaculture and marketed as live, fresh, and frozen due to its high economic value as food fish [2]. It can live in a variety of low-oxygen settings, including swamps and marshes, and burrows within the mudflat throughout the summer [3]. Adaptations like terrestrial dispersal, aerial respiration, and high tolerance to hypoxia and ammonia can therefore be studied using *C. batrachus* as the ideal model [4].

Fish is considered as a crucial constituent of supportable diets for the future [5]. Subsequently, fish production stagnates, upcoming demand will depend on aquacultural products [6]. Growing aquaculture production will need an upsurge production of fish feed [7] and to improve the capability of farmed fish to assimilate feed consumption into biomass which can help to reduce feed use in the fisheries industry and thus will enhance its sustainability through condensed expenditures and ecological effects [8].

An important issue in fisheries industry is feeding, particularly when it affects production costs and the health and growth of fish [9]. Feed expenses mark up 40–50% of total production costs of fish [10]. Furthermore, growth optimization in the fish farming system is imperative to confirm success [11]. Fish growth at all stages in its life history is mainly controlled by different factors, including feeding rate, food intake, feeding frequency, food type, and the ability for nutrient absorption [12]. The capability to alter ingested feed into body mass growth can be observed by the feed conversion ratio (FCR). It can be enhanced by modification in feed composition [13]. As, FCR is a commonly used measure of conversion which is commonly used over the entire production life of a fish, and obviously, the amount of feed changes during this period [14], however, assessing the FCR in different life stages, like juvenile and younger stages, may be helpful for better management.

In fish feed, protein is the largest nutrient and considered the most costly source for quality health and optimum growth of a fish [15], but it also influences growth performance and feeds conversion ratio of fish [16–18]. Fish usually ingest protein to attain non-essential and essential amino acids, essential for enzymatic function, muscle formation, and to supply energy [19]. Though, both excessive and inadequate protein in the feed not only affects the quality and growth of fish but also influence expenditure on aquaculture and as well as water quality. Therefore, optimal dietary protein level is imperative for best growth and to support good health in fish culture system [20].

Proximate body composition helps to rank different fish species based on their nutritional and functional benefits and to assess the energy value of the fishes. Thus, allows consumers, feed formulators and researchers to select the fish according to their requirements for their nutritional values and/or processing [21–24]. The significance of proximate composition has been discovered in the study of fish bioenergetics and the effect of pollutants. It acts as a good indicator of fish physiology [25]. Fish proximate body composition constituents (fat, protein, water, organic content and ash) are influenced by diet, feed rate, sex, genetic strain, age, species and also by changing body size and condition factor [26–28].

The objective of the present study was to assess the survival rate, growth performance, feed conversion ratio and proximate composition of *Clarias batrachus* fed on varying dietary protein levels.

## 2. Materials and methods

### 2.1. Ethics statement

**2.1.1. Institutional review board statement.** It has been confirmed that the experimental data collection complied with appropriate permissions from Ethical Review Committee,

Department of Zoology, Faculty of Biological Sciences, Quaid-i-Azam University, Islamabad, Pakistan. The study did not involve humans.

## 2.2. Experimental design

Fish fry of *Clarias batrachus* comprising 0.5–1.38 g wet body weight (W) and 3.90–7.40 cm total length (TL) were procured from the Tawakkal Fish Hatchery & Farm, Muzaffargarh, Pakistan, and acclimatized on rice polish in glass aquaria for two weeks. Feeding trial was conducted in eighteen glass aquaria (volume 50 L), each having working dimensions of 60 x 40 x 44 cm$^3$. After acclimatization, a total of 180 fish fry of *C. batrachus* were randomly stocked comprising ten fish in each aquarium (30 per treatment) and three replicates were followed for each treatment.

Six experimental diets (Table 1) comprising 25, 30, 35, 40, 45, and 50% crude protein (CP) were supplied to the fish in treatment-1 (T1), treatment-2 (T2), treatment-3 (T3), treatment-4 (T4), treatment-5 (T5) and treatment-6 (T6), respectively. Proximate composition of experimental feeds was calculated following the studies of NRC [29], Preston [30] and NDDB [31]. Feed was given to the fish at the rate of 5% of fish body weight for the entire 90 days of the experimental period, twice (0900 and 1800) in two equal meals, with 16 hours light and 8 hours dark cycle daily. Dissolved oxygen and pH of water in each aquarium were monitored daily. The temperature of each aquarium was maintained at 24–26°C during the study period. Size (W and TL) of each stocked *C. batrachus* were recorded after 15 days to adjust the feeding rates and to calculate different growth parameters i.e. length gain, percent length gain, mean final weight, weight gain, daily growth rate, percent weight gain, feed conversion ratio (FCR),

**Table 1. Ingredients (%) used for feed formulation and proximate composition of various diets.**

| Ingredients | T1 (CP-25) | T2 (CP-30) | T3 (CP-35) | T4 (CP-40) | T5 (CP-45) | T6 (CP-50) |
|---|---|---|---|---|---|---|
| Canola Meal | 5 | 5 | 5 | 5 | 5 | 2 |
| Corn Gluten Meal 30% | 10 | 9 | 5 | 5 | 3 | 0 |
| Corn Gluten Meal 60% | 10 | 15 | 15 | 15 | 5 | 3 |
| Fishmeal | 10 | 10 | 15 | 21 | 30 | 37 |
| Rice Polish | 25 | 15 | 10 | 4 | 3 | 0 |
| Sarson Meal | 5 | 2 | 5 | 5 | 3 | 5 |
| Soybean Meal | 10 | 15 | 25 | 31 | 40 | 44 |
| Sunflower Meal | 5 | 5 | 5 | 5 | 3 | 2 |
| Wheat Bran | 15 | 19 | 10 | 4 | 3 | 2 |
| Dicalcium Phosphate | 1 | 1 | 1 | 1 | 1 | 1 |
| Vitamin Premixes | 1 | 1 | 1 | 1 | 1 | 1 |
| Soybean Oil | 2 | 2 | 2 | 2 | 2 | 2 |
| Carboxymethyl Cellulose (CMC) | 1 | 1 | 1 | 1 | 1 | 1 |
| **Proximate Composition of the Diets (%)** | | | | | | |
| Moisture | 8.96 | 9.28 | 8.95 | 8.25 | 8.86 | 8.9 |
| Dry Matter (DM) | 91.04 | 90.72 | 91.05 | 91.75 | 91.14 | 91.1 |
| Crude Protein (CP) | 25.15 | 29.95 | 30.00 | 34.80 | 40.10 | 45.03 |
| Crude Fat (CF) | 8.11 | 6.91 | 7.20 | 7.50 | 7.82 | 8.02 |
| Ash | 8.16 | 7.83 | 7.63 | 7.86 | 7.90 | 7.91 |
| Fiber | 8.38 | 8.05 | 8.10 | 8.03 | 8.22 | 8.07 |
| Nitogen Free Extract (NFE) | 41.24 | 37.98 | 38.12 | 33.56 | 27.10 | 22.07 |

NFE = DM-(%CP+ %CF + %Ash + %Fiber)

specific growth rate (SGR%) and protein efficiency ratio (PER). At the end of the feeding trial, the total number of fingerlings in each tank was counted to calculate survival rate.

Growth performance of *C. batrachus* fed varying levels of dietary protein was measured as a function of the weight gain by calculating the following statistics:

Percent Length Gain (%LG) = (Avg. final length− Avg. initial length /initial length) × 100

Percent weight gain (%WG) = (Avg. final weight − Avg. initial weight/initial weight) × 100

Daily Growth Rate (DGR) = Avg. final body weight − Avg. initial body weight/growth period (in days)

Feed conversion ratio (FCR) = Dry feed intake (g)/biomass gain (g)

Specific growth rate (SGR%) = 100× (Ln final weight − Ln initial weight) / growth period

Protein efficiency ratio (PER) = Weight gain (g)/protein intake (g)

At the end of the feeding trial, fish specimens were immersed and kept in solution of MS222 (250mg/L) for 10 minutes to euthanize the fish. Proximate composition was analysed by taking of whole body of the fish. In brief, specimens of *C. batrachus* were dried to a constant weight at 80˚C in an oven (Incucell, MMM Medcenter Einrichtungen GmbH, MMM-Group) to determine water content in the fish. Ash was determined by incineration using muffle furnace (RJM-1.8-10A) at 550˚C for 12 hr. Fat was determined by extracting in a chloroform and methanol solution (1:2). Protein contents in *C. batrachus* were assessed by difference from mass of other constituents, following the approach adopted by Naeem and Salam [27].

## 2.3. Statistical analyses

The data were subjected to ANOVA followed by Duncan's new multiple range test to study the differences among treatments in SPSS version 23. Mean differences among treatment were determined by Duncan's multiple range test and considered significant at $p < 0.05$. Correlation and regression analyses (Y = a + bX) were also performed to study the effect of fish size on proximate composition in *C. batrachus*. Correlation coefficients for regression analyses were considered significant at $p < 0.05$, $p < 0.01$ and $p < 0.001$.

## 3. Results

Growth performance of the walking catfish (*Clarias batrachus*) fed with different feed treatments (25%, 30%, 35%, 40%, 45%, and 50% CP) is presented in Table 2. Survival rate (%) of *C. batrachus* was found 100% in all the studied treatment groups. Fortnightly length gain (cm) and fortnightly weight gain (g) of *C. batrachus* in different studied treatments is represented in Figs 1 and 2, respectively.

Length gain and percent length gain (%LG) of *C. batrachus* showed non-significant differences ($P > 0.05$) among various studied treatment groups. However, dietary protein levels significantly affected ($P < 0.05$) mean final weight, weight gain, daily growth rate, percent weight gain, FCR, SGR%, and PER of *C. batrachus* (Table 2*)*.

Significantly highest ($P < 0.001$) mean value (6.96±0.04) of weight gain (g/fish) in *C. batrachus* was recorded in the fishes that were reared in T4 by feeding 40%CP in the diet. Daily growth rate of *C. batrachus* was significantly highest ($P < 0.05$) in T4 (40%CP) with a mean (±SE) value of 0.63±0.005. The highest ($P < 0.001$) mean (±SE) value of percent weight gain (80.84±0.28%) was also recorded in the fishes that were fed 40%CP in T4, while that was the lowest (56.14±1.04) in T1 (25%CP), as shown in Table 2.

The best FCR value of *C. batrachus* was obtained in T4 by feeding the fish 40% crude protein in diet, as it remained the significantly ($P < 0.001$) lowest (1.90±0.02) among various studied treatments. While the highest FCR value was recorded as 5.24±0.11 in T1, in which *C. batrachus* were supplied 25% dietary protein level. Specific growth rate (SGR%) of *C. batrachus*

**Table 2. Descriptive statistics (Mean±SE) for various growth parameters of *Clarias batrachus* fed upon various dietary crude protein levels in different treatments.**

| Parameters | T1 (CP-25) | T2 (CP-30) | T3 (CP-35) | T4 (CP-40) | T5 (CP-45) | T6 (CP-50) | p-value |
|---|---|---|---|---|---|---|---|
| Survival rate of fish (%) | 100 | 100 | 100 | 100 | 100 | 100 | — |
| Mean Initial Length (cm) (iTL) | 5.29±0.04 ns | 5.25±0.02 ns | 5.28±0.02 ns | 5.31±0.05 ns | 5.38±0.08 ns | 5.27±0.09 ns | .692 |
| Mean Final Length (cm) (fTL) | 6.71±0.08 ns | 6.59±0.03 ns | 6.72±0.08 ns | 6.67±0.09 ns | 6.70±0.10 ns | 6.60±0.10 ns | .793 |
| Length gain (cm) (LG) | 1.42±0.06 ns | 1.32±0.04 ns | 1.44±0.07 ns | 1.36±0.04 ns | 1.33±0.02 ns | 1.33±0.07 ns | .439 |
| Percent Length Gain (%LG) | 21.15±0.63 ns | 19.96±0.57 ns | 21.42±0.78 ns | 20.37±0.30 ns | 19.79±0.03 ns | 20.16±0.93 ns | .384 |
| Mean Initial Weight (g) (iW) | 1.37±0.01 ns | 1.30±0.01 ns | 1.37±0.02 ns | 1.33±0.01 ns | 1.33±0.02 ns | 1.31±0.03 ns | .093 |
| Mean Final Weight (g) (fW) | 3.13±0.04$^c$ | 3.26±0.03$^c$ | 5.41±0.01$^b$ | 6.96±0.04$^a$ | 5.01±0.01$^b$ | 4.18±0.05$^c$ | < .001 |
| Weight Gain (g/fish) (WG) | 1.76±0.06$^c$ | 1.96±0.02$^c$ | 4.04±0.01$^b$ | 5.63±0.05$^a$ | 3.68±0.04$^b$ | 2.87±0.04$^c$ | < .001 |
| Percent Weight Gain (%WG) | 56.14±1.04$^c$ | 60.18±0.22$^c$ | 74.63±0.35$^b$ | 80.84±0.28$^a$ | 73.40±0.54$^b$ | 71.97±0.47$^b$ | < .001 |
| Daily Growth Rate (g/day) (DGR) | 0.19±0.004$^c$ | 0.22±0.002$^c$ | 0.45±0.002$^b$ | 0.63±0.005$^a$ | 0.41±0.004$^b$ | 0.32±0.004$^{bc}$ | < .001 |
| Feed Conversion Ratio (FCR) | 5.24±0.11$^a$ | 4.78±0.012$^a$ | 2.70±0.02$^b$ | 1.90±0.02$^c$ | 2.64±0.04$^b$ | 3.33±0.09$^b$ | < .001 |
| Specific Growth Rate (SGR%) | 0.40±0.01$^c$ | 0.44±0.002$^c$ | 0.66±0.01$^b$ | 0.80±0.01$^a$ | 0.64±0.01$^b$ | 0.55±0.01$^{bc}$ | < .001 |
| Protein Efficiency Ratio (PER) | 0.76±0.02$^{bc}$ | 0.70±0.002$^c$ | 1.06±0.01$^b$ | 1.32±0.01$^a$ | 0.84±0.01$^b$ | 0.60±0.02$^c$ | < .001 |

Mean values sharing the same superscript in a row are not significantly different ($p>0.05$), ns = not significant

T1 = 25%CP, T2 = 30%CP, T3 = 35%CP, T4 = 40%CP, T5 = 45%CP, T6 = 50%C

was found significant highest (P<0.05) in T4(40%CP) with mean value being 0.80±0.01, while the lowest (0.40±0.01) in T1 by providing fish 25% crude protein in diet. Highest (P<0.05) mean value of protein efficiency ratio (PER) for *C. batrachus* was also recorded as 1.32±0.01in feeding treatment which was supplied with a diet containing 40% protein level (T4). The lowest mean PER value (0.60±0.02) for *C. batrachus* was found in T6 in which fish were supplied a diet containing 50% protein.

Mean value of water, ash, fat and protein contents (% wet mass) were ranged from 74.10 ±0.31% - 79.23±0.52%, 3.12±0.07% - 4.68±0.05%, 3.90±0.06% - 4.43±0.05% and 13.09±0.58% -

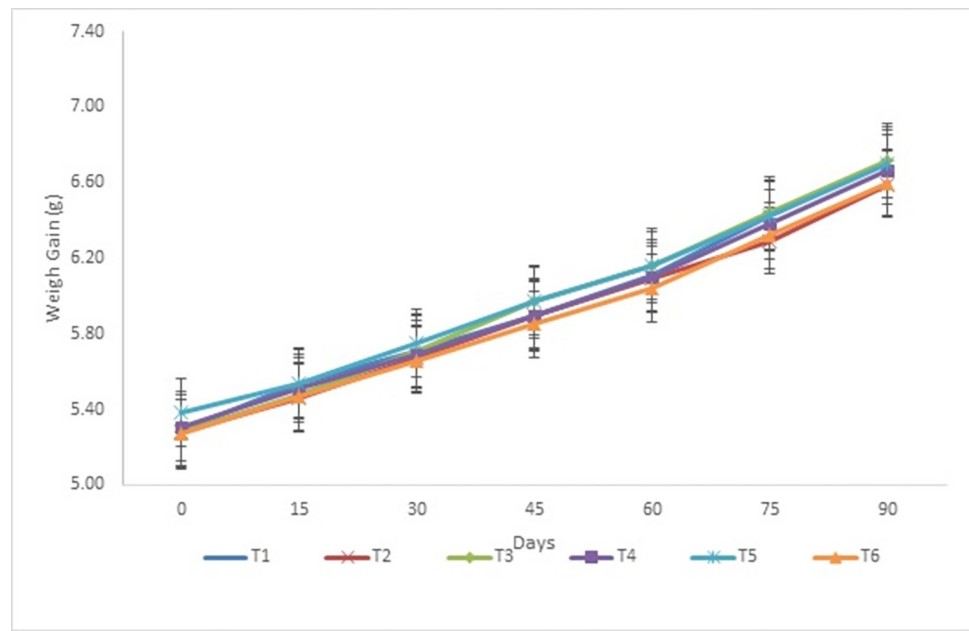

**Fig 1. Fortnightly mean length (cm) of *Clarias batrachus* in different studied treatments.**

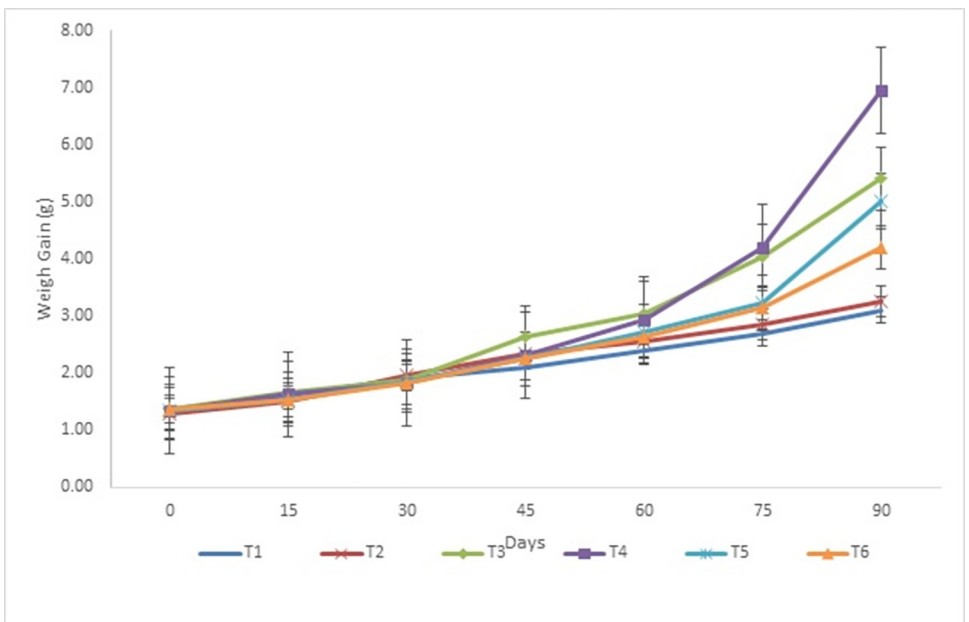

**Fig 2. Fortnightly mean weight (g) of *Clarias batrachus* in different studied treatments.**

16.79±0.27% in the studied treatments (Table 3). Analysis of variance (ANOVA) revealed significant difference in all body constituents (both in wet and dry masses) among the studied treatments (T1-T6). Water content (%) was found significantly ($p<0.05$) higher in the body of *C.batrachus* for T1, T2, T3 and T6 than for T4 and T5. Ash (wet mass) was found significantly higher in the fish reared in T4 (4.68±0.05%) and T5 (4.53±0.06%), while the lowest in T6 (3.12 ±0.07%). Fat content in the wet body mass of *C. batrachus* was found significantly higher in T4 and T1. Significant higher value of mean protein content in wet mass was noted in *C. batrachus* reared in T4 (CP-40) and T5 (CP-45).

Water content (%) in the body of *C. batrachus* showed highly significant negative correlation ($p<0.001$) with ash content (% wet mass) reared in T1 by feeding 25% of crude protein, and significant ($p< 0.01$) in T2, T3 and T6; while non-significant correlation was observed in T4 and T5. Fat content was found significantly correlated ($p<0.001$) with water (%) only in T1 and T2. However, protein content of ($p<0.001$) showed highly significant negative correlation ($p<0.001$) in all the studied treatment (Table 4).

**Table 3. Mean values (± SE) of various constituents in percentage (%) of wet and dry mass of *Clarias batrachus* reared in different treatments.**

| Constituents | | T1 (CP-25) | T2 (CP-30) | T3 (CP-35) | T4 (CP-40) | T5 (CP-45) | T6 (CP-50) | p-value |
|---|---|---|---|---|---|---|---|---|
| Water (%) | | 78.81±0.74[a] | 79.23±0.52[a] | 78.33±0.30[a] | 74.10±0.31[b] | 75.50±0.50[b] | 78.97±0.40[a] | < .001 |
| Ash | Wet mass | 3.68±0.14[b] | 3.72±0.10 [b] | 3.41±0.09[c] | 4.68±0.05[a] | 4.53±0.06[a] | 3.12±0.07[d] | < .001 |
| | Dry mass | 17.50±0.43 [b] | 18.03±0.45[ab] | 15.73±0.34 [c] | 18.14±0.25 [ab] | 18.69±0.43 [a] | 14.90±0.31[c] | < .001 |
| Fat | Wet mass | 4.42±0.11 [a] | 3.95±0.09 [c] | 3.90±0.06[c] | 4.43±0.05 [a] | 4.18±0.07 [b] | 4.22±0.07[ab] | .002 |
| | Dry mass | 21.22±0.52[a] | 19.15±0.36 [b] | 18.09±0.34[bc] | 17.13±0.21[c] | 17.28±0.51[c] | 20.28±0.53 [ab] | .001 |
| Protein | Wet mass | 13.09±0.58 [b] | 13.09±0.41 [b] | 14.35±0.26[b] | 16.79±0.27 [a] | 15.80±0.51[a] | 13.70±0.40 [b] | < .001 |
| | Dry mass | 61.29±0.83[c] | 62.82±0.62[bc] | 66.18±0.55 [a] | 64.73±0.38 [ab] | 64.03±0.86 [b] | 64.82±0.76[ab] | < .001 |

Mean values sharing the same superscript in a row are not significantly different ($p>0.05$)

**Table 4. Statistical regression parameters of percentage water (%W) content versus % body constituents in wet mass of *C. batrachus* reared in different treatments.**

| Equation | Treatment | r | a | b | SE of b | t-Stat |
|---|---|---|---|---|---|---|
| %Ash = a + b %Water | T1 | -0.754*** | 14.631 | -0.139 | 0.023 | -5.965 |
| | T2 | -0.563** | 12.657 | -0.113 | 0.032 | -3.538 |
| | T3 | -0.552** | 15.843 | -0.159 | 0.046 | -3.439 |
| | T4 | -0.324[ns] | 8.729 | -0.055 | 0.031 | -1.783 |
| | T5 | -0.144 [ns] | 5.839 | -0.017 | 0.023 | -0.758 |
| | T6 | -0.416** | 8.506 | -0.068 | 0.029 | -2.382 |
| %Fat = a + b % Water | T1 | -0.703*** | 12.654 | -0.105 | 0.020 | -5.143 |
| | T2 | -0.702*** | 13.442 | -0.120 | 0.023 | -5.123 |
| | T3 | -0.266 [ns] | 8.139 | -0.054 | 0.038 | -1.434 |
| | T4 | -0.324 [ns] | 8.729 | -0.055 | 0.031 | -1.783 |
| | T5 | 0.081 [ns] | 3.293 | 0.012 | 0.028 | 0.424 |
| | T6 | 0.105 [ns] | 2.702 | 0.019 | 0.035 | 0.549 |
| % Protein = a + b % Water | T1 | -0.967*** | 72.715 | -0.757 | 0.039 | -19.600 |
| | T2 | -0.965*** | 73.900 | -0.767 | 0.040 | -18.993 |
| | T3 | -0.909*** | 76.017 | -0.787 | 0.069 | -11.345 |
| | T4 | -0.968*** | 80.655 | -0.862 | 0.043 | -20.081 |
| | T5 | -0.982*** | 90.869 | -0.994 | 0.037 | -26.654 |
| | T6 | -0.959*** | 88.793 | -0.951 | 0.054 | -17.500 |

r = Correlation Coefficient; a = Intercept; b = Slope; S.E = Standard Error

*** = $p < 0.001$

** = $p < 0.01$; [ns] $> 0.05$

Size (wet weight and total length) of the fish represented highly significant correlation ($p < 0.001$) with all the body constituents in all the studied treatment groups for the studied catfish, *C. batrachus* (Tables 5 and 6).Water showed negative allometric pattern for all the studied treatments groups with an increase in body weight of *C. batrachus*. Ash contents showed positive allometry for all the studied treatments groups except for T3 (CP35) which represented isometric pattern (b = 1.036). Slope (b value) represented positive allometry for fats in the body of *C. batrachus* in T1, T2 and T4, negative allometry in T6, while isometry in T3 and T5. Positive allometry was also found in all treatments except for T5 (CP45) which represented isometry with an increase in body weight of the studied fish (Table 5). On the other hand, all the body constituents showed negative allometry for all studied treatment groups with an increase in total length of *C. batrachus* (Table 6).

Table 7 showed that condition factor remained significantly correlated with water only in T2 (r = 0.427, $p < 0.01$), T3 (r = 0.618, = $p < 0.001$) and T4 (r = 0.540, $p < 0.01$). Ash content was found insignificantly correlated ($p > 0.05$) in all treatments except for T2 which was found negatively correlated (r = 0.608, $p < 0.001$) with condition factor of *C. batrachus*. Fat was also found significant (r = 0.367, p < 0.05) with condition factor for only T4, in which the fish was fed a diet containing 40% crude protein. Significant correlation was also found in T3 (r = 0.636, $p < 0.01$) and T4 (r = 0.507, $p < 0.001$).

## 4. Discussion

Evaluation of the optimum dietary protein level is one of the most critical factors for the success of aquaculture operations. In the present work growth performance of *Clarias batrachus* was compared between different feeding treatments with six varying levels of formulated diets

**Table 5. Statistical regression parameters of log transformed wet body weight (g) versus log transformed total body constituents in wet mass of *C. batrachus* reared in different treatments.**

| Equation | Treatment | r | a | b | S. E. (b) | *t* value when b = 1 |
|---|---|---|---|---|---|---|
| Water = *a* + *b* Wet Weight | T1 | 0.983*** | 0.019 | 0.746 | 0.027 | -36.23 |
| | T2 | 0.984*** | -0.005 | 0.811 | 0.028 | -34.54 |
| | T3 | 0.992*** | -0.018 | 0.879 | 0.022 | -45.01 |
| | T4 | -0.968*** | 80.655 | -0.862 | 0.043 | -20.081 |
| | T5 | 0.957*** | 0.079 | 0.712 | 0.042 | -23.28 |
| | T6 | 0.971*** | -0.017 | 0.863 | 0.041 | -23.65 |
| Ash = *a* + *b* Wet Weight | T1 | 0.960*** | -1.906 | 1.956 | 0.109 | -7.18 |
| | T2 | 0.830*** | -1.637 | 1.399 | 0.181 | -4.13 |
| | T3 | 0.680*** | -1.498 | 1.036 | 0.215 | -3.62 |
| | T4 | 0.798*** | -1.450 | 1.143 | 0.166 | -4.87 |
| | T5 | 0.841*** | -1.432 | 1.124 | 0.139 | -6.05 |
| | T6 | 0.845*** | -1.798 | 1.465 | 0.178 | -4.14 |
| Fat = *a* + *b* Wet Weight | T1 | 0.963*** | -1.661 | 1.624 | 0.088 | -9.80 |
| | T2 | 0.927*** | -1.676 | 1.530 | 0.119 | -6.87 |
| | T3 | 0.817*** | -1.400 | 0.987 | 0.134 | -6.47 |
| | T4 | 0.905*** | -1.849 | 1.587 | 0.143 | -5.38 |
| | T5 | 0.718*** | -1.369 | 0.983 | 0.183 | -4.48 |
| | T6 | 0.676*** | -1.233 | 0.769 | 0.161 | -5.43 |
| Protein = *a* + *b* Wet Weight | T1 | 0.916*** | -1.410 | 2.062 | 0.173 | -3.70 |
| | T2 | 0.921*** | -1.304 | 1.818 | 0.148 | -4.93 |
| | T3 | 0.959*** | -1.313 | 1.641 | 0.093 | -9.09 |
| | T4 | 0.877*** | -1.482 | 1.838 | 0.194 | -3.33 |
| | T5 | 0.718*** | -1.369 | 0.983 | 0.183 | -4.48 |
| | T6 | 0.820*** | -1.325 | 1.736 | 0.233 | -2.56 |

r = Correlation Coefficient; a = Intercept; b = Slope; S.E = Standard Error

*** = $p < 0.001$

containing 25% (T1), 30% (T2), 35% (T3), 40% (T4), 45% (T5) and 50% crude protein (T6). Results of the growth experiment showed that the survival rate was recorded as 100% in all of the studied groups, indicating high tolerance of the fish in the confined system and also represents that rearing conditions were good (optimal). Results of the present study agreed well with a previously conducted study by Farhat and Khan [32], which reported a survival rate of 100% in *C. gariepinus* by feeding the fish with 30%, 35%, 40%, 45%, and 50% crude protein (CP).

Results of present study also revealed that highest daily growth rate (0.63±0.005 g/day) was observed in fish that were supplied 40% CP (T4) followed by 35% CP (0.45±0.002 g/day) in T3, and the lowest (0.19±0.004 g/day) was found in fish fed on 25% CP in T1 for *C. batrachus*. The observed difference in daily growth rate among the treatments was found to be statistically ($p < 0.05$) significant. Treatment groups showed difference in growth rate indicating the importance of supplementary feeds on the growth and production of fish. In general, the daily growth rate of *C. batrachus* recorded in the present experiment was found similar to that previously described by Tadesse [33], who have reported a daily growth rate of 0.23 to 0.52 g/day in African catfish, *Clarias gariepinus*, fingerlings which were reared in tanks; but lower than that of documented (1.12 g/day to 1.64 g/day) by Yalew [34], for different stocking density in *C. gariepinus*. This lower growth rate of the fish than the study of Yalew [34] might be due to the difference in feed composition of the ingredients or difference in fish size.

**Table 6. Statistical regression parameters of log transformed total length (cm) versus log transformed total body constituents (g) in wet mass of *C. batrachus* reared in different treatments.**

| Equation | Treatment | r | a | b | S. E. (b) | *t* value when b = 1 |
|---|---|---|---|---|---|---|
| Water = *a* + *b* Total Length | T1 | 0.664*** | -0.412 | 0.961 | 0.208 | -13.48 |
| | T2 | 0.801*** | -0.450 | 1.048 | 0.151 | -18.84 |
| | T3 | 0.802*** | -0.029 | 0.792 | 0.113 | -25.65 |
| | T4 | 0.794*** | 0.320 | 0.476 | 0.070 | -42.32 |
| | T5 | 0.854*** | 0.054 | 0.633 | 0.074 | -39.87 |
| | T6 | 0.836*** | -0.146 | 0.814 | 0.103 | -28.43 |
| Ash = *a* + *b* Total Length | T1 | 0.662*** | -3.078 | 2.568 | 0.559 | -2.80 |
| | T2 | 0.864*** | -2.817 | 2.313 | 0.259 | -9.25 |
| | T3 | 0.675*** | -1.687 | 1.147 | 0.241 | -11.31 |
| | T4 | 0.686*** | -1.096 | 0.739 | 0.151 | -19.16 |
| | T5 | 0.749*** | -1.469 | 0.997 | 0.170 | -16.69 |
| | T6 | 0.782*** | -2.101 | 1.486 | 0.228 | -11.69 |
| Fat = *a* + *b* Total Length | T1 | 0.660*** | -2.623 | 2.119 | 0.464 | -4.35 |
| | T2 | 0.780*** | -2.571 | 2.046 | 0.316 | -7.46 |
| | T3 | 0.704*** | -1.462 | 0.949 | 0.184 | -15.36 |
| | T4 | 0.686*** | -1.096 | 0.739 | 0.151 | -19.16 |
| | T5 | 0.631*** | -1.391 | 0.859 | 0.203 | -13.91 |
| | T6 | 0.586*** | -1.352 | 0.730 | 0.194 | -14.72 |
| Protein = *a* + *b* Total Length | T1 | 0.588*** | -2.492 | 2.521 | 0.666 | -1.98 |
| | T2 | 0.767*** | -2.347 | 2.406 | 0.387 | -5.34 |
| | T3 | 0.845*** | -1.444 | 1.611 | 0.196 | -13.67 |
| | T4 | 0.748*** | -0.905 | 1.179 | 0.201 | -13.74 |
| | T5 | 0.606*** | -1.415 | 1.583 | 0.399 | -5.93 |
| | T6 | 0.697*** | -1.566 | 1.615 | 0.320 | -7.77 |

r = Correlation Coefficient; a = Intercept; b = Slope; S.E = Standard Error

*** = $p < 0.001$

FCR (Feed conversion ratio) is a vital gauge of the fineness of fish diet. A lower FCR designates better consumption of feed by a fish [35]. In the present research, the FCR values ranged from 1.90 to 5.24 and varied significantly ($p < 0.05$) between feeding treatments. Ogunji et al. [36] declared FCR values of 1.2–1.5 as good range for fish raised with balanced diet and also suggested that inclusion of more animal ingredients in fish diet may provide higher growth rate and lower FCR of fish in the future. Although **FCR** value between 1.2 and 1.5 represent a good indicator, it could not be used as absolute standard in all fish species, as it can change according to several culturing factors. As, Tadesse [33] reported the lowest feed conversion ratio (FCR = 2.14) in fish fed with 40% CP, indicating its suitability for African catfish fingerlings than the other test diets. Further, in the present investigation, the lowest FCR value (1.90 ±0.02) was noted in fish provided with 40% CP, representing that the fish consumed the feed better than the other test feeds (25%, 30%, 35%, 45%, 50% CP) of the experiment for *C. batrachus*. The best FCR for *C. batrachus* in T4 might be due to the high proportion of protein derived from easily digestible animal ingredients. FCR of the present study in *C. batrachus* is very similar to those reported by Tadesse [33]. On the other hand, Ogunji and Awoke [37], have reported lower FCR values of 1.61 for *C. gariepinus* reared in tanks under greenhouse. Unlike these results of FCR in catfishes, Iqbal and Naeem [18], and Ishtiaq and Naeem [16], noted the lowest food conversion ratio (FCR) fed upon 25% CP in the carp hybrid fry (*Labeo*

**Table 7. Statistical regression parameters of condition factor (K) versus percentages (%) of body constituents (wet mass, g) for *C. batrachus* reared in different treatments.**

| Equation | Treatment | r | a | b | S. E. (b) | t value when b = 1 |
|---|---|---|---|---|---|---|
| Water = a + b Condition Factor | T1 | 0.019[ns] | 78.436 | 0.364 | 3.600 | 0.101 |
| | T2 | 0.427** | 72.967 | 5.417 | 2.204 | 2.457 |
| | T3 | 0.618*** | 74.220 | 2.211 | 0.541 | 4.085 |
| | T4 | 0.540** | 70.019 | 1.685 | 0.505 | 3.338 |
| | T5 | 0.303[ns] | 72.029 | 2.010 | 1.218 | 1.651 |
| | T6 | 0.351[ns] | 75.625 | 2.203 | 1.131 | 1.948 |
| Ash = a + b Condition Factor | T1 | -0.068[ns] | 3.923 | -0.235 | 3.593 | -0.065 |
| | T2 | -0.608*** | 5.509 | -1.546 | 1.935 | -0.799 |
| | T3 | -0.203[ns] | 3.798 | -0.209 | 0.674 | -0.311 |
| | T4 | -0.151[ns] | 4.876 | -0.079 | 0.593 | -0.133 |
| | T5 | -0.235[ns] | 4.853 | -0.188 | 1.242 | -0.151 |
| | T6 | -0.400* | 3.741 | -0.411 | 1.107 | -0.371 |
| Fat = a + b Condition Factor | T1 | -0.122[ns] | 4.768 | -0.338 | 3.574 | -0.095 |
| | T2 | -0.317[ns] | 4.743 | -0.685 | 2.313 | -0.296 |
| | T3 | -0.041[ns] | 3.960 | -0.030 | 0.688 | -0.043 |
| | T4 | -0.367* | 4.907 | -0.199 | 0.558 | -0.356 |
| | T5 | -0.016[ns] | 4.202 | -0.015 | 1.277 | -0.012 |
| | T6 | 0.200[ns] | 3.868 | 0.230 | 1.183 | 0.194 |
| Protein = a + b Condition Factor | T1 | 0.014[ns] | 12.874 | 0.209 | 3.601 | 0.058 |
| | T2 | -0.316[ns] | 16.781 | -3.186 | 2.313 | -1.377 |
| | T3 | -0.636*** | 18.023 | -1.971 | 0.531 | -3.713 |
| | T4 | -0.507** | 20.198 | -1.407 | 0.517 | -2.721 |
| | T5 | -0.269[ns] | 18.916 | -1.808 | 1.231 | -1.469 |
| | T6 | -0.324[ns] | 16.766 | -2.022 | 1.143 | -1.769 |

r = Correlation Coefficient; a = Intercept; b = Slope; S.E = Standard Error

*** = $p < 0.001$

** = $p < 0.01$

* = $p < 0.05$; [ns]$p > 0.05$

*rohita* ♀ and *Catla catla* ♂) and carp (*Catla catla*), respectively. While Khalid and Naeem [38], reported lowest FCR in grass carp (*Ctenopharyngodon idella*) by feeding only 20% protein in diet. The variation might be due to differences in dietary habits of carps and catfishes.

In the present investigation, the SGR value was highest for *C. batrachus* fed with 40% protein and lowest for 25% dietary protein in fish. SGR increases with increasing dietary protein levels up to 40% in *C. batrachus* and above optimum protein level, SGR decreased. These results agree with the outcomes of Mohanta et al. [39] and Gandotra et al. [40] who also reported that SGR increased with increasing dietary protein levels.

Present observation further reveals that fish-fed diet having 40% protein displayed significantly (P<0.05) higher PER than those supplied with other dietary protein levels. Though a propensity of growing PER values from 0.60 to 1.32 with the increase of each protein level in diet up to 40% was noted, and afterward a significant drop in PER values were recorded in diets containing higher protein level. Similar trend was also documented by Ahmed and Ahmad [17], in *Oncorhynchus mykiss* fingerlings reared in the Himalayan region of India.

In the present study, whole body proximate composition of *C. batrachus* was categorically affected by dietary treatment. Similar results were recorded for *Pagrus pagrus*, [41], *Totoaba*

*macdonaldi* [42], *Culter alburnus* Basilewsky [43], *Catla catla* [44], and Genetically Improved Farmed Tilapia [45]. It is also documented that proximate composition depends on the fish species, fish size, dietary protein sources and environmental conditions [46]. However, this finding is in contrast to those reported in other studies for *Seriola dumerili* [47] and *Solea senegalensis* [48] and *Siniperca scherzeri* [49].

Present study revealed that mean value of water and protein contents were ranged 74.10–79.23% and 13.09–16.79% in the studied treatment in which *C. batrachus* were fed with different levels of dietary protein. These values fit within the range of those reported in previous study for other strictly carnivorous fish species of the same genus, *Clarias gariepinus* [50]. Significantly lower water content and higher whole-body protein contents were found in *C. batrachus* fed with a diet containing 40% CP and 45% CP than those of the fish fed with 25%, 30%, 35% and 50% CP in diets. However, Kim et al. [51] have reported no significant effect of dietary protein levels of 20%, 30% and 40% on crude protein of the Juvenile Catfish, *Silurusasotus*. On the other hand, Ishtiaq and Naeem [44] have observed that dietary crude protein levels definitely affect the water and protein contents of *Catla catla*. Hence, the results indicated that farmer can achieve not only higher growth and low FCR but can attain higher protein and lipid contents in the *C. batrachus* by feeding the fish with 40% crude protein, rather higher crude protein (45%CP) in diet.

Ash contents of *C. batrachus* fed with the experimental diets was also significantly affected by dietary protein levels, which is in accordance with Hien et al. [52] for *Clarias microcephalus*. The body lipid content generally increased as the dietary protein level increased in the present investigation as previously reported by Bai et al. [53] for yellow puffer. On the contrary, Kim et al. [51] documented that as the protein content of whole body increases, whole-body lipid content decreases. However, in the present study, though, lipid contents were significantly affected by dietary protein levels, but no increasing or decreasing effect was observed with an increase in dietary protein. These discrepancies may be attributed to the difference in experimental condition mainly dietary protein levels or due to fish species variation, as feed, intensive feeding or starvation, maturity stage [54, 55], sex [56], condition factor [57], age and seasonal variations [58] and body size [59] have been found to have a pronounced impact on the proximate composition of different fish species.

Literature shows that body weight of a fish influences the various body constituents [16], and fat and protein increase while ash and water decrease with the increasing total length [59]. Hence, regression analyses were also performed to observe the effect of fish size on proximate composition of *C. batrachus*. Highly significant correlation ($p<0.001$) between fish size (weight and length) and body constituents (water, ash, fat and protein) were noted in the present work and found in agreement to those reported by Naeem and Ishtiaq [60] for *Mystus bleekeri*, Bano et al. [61] for *Labeo calbasu* and Ishtiaq and Naeem [16] for *Catla catla*. Negative allometric pattern for water contents indicated an increase in this content with lesser proportion. While protein constituents were found increasing with greater proportion (positive allometrric pattern) with an increase in fish size when compared with b = 1 for body weight and b = 3 for total length. The negative allometry for water and positive allometry for protein is evident in many studies [16, 43, 60, 61] and hence the findings of present study indorse the same trend.

Furthermore, body composition of a fish species can be assessed from water content by performing regression analyses. It allows to predict other constituents of body (protein, fat and ash), with the consistent lessening of costs when performing one, in spite of diverse analyses. These opinions have furnished by [16, 62], and are in agreement with the results of the present study, especially for protein contents in the body of *C. batrachus* which showed negative correlation ($p<0.001$) in all the studied treatments, which is found in in conformity with the findings of those reported by Bano et al. [61].

Some studies documented a noticeable impact of condition factor on the proximate composition, however, most of the studies have reported insignificant relationships between body constituents and condition factor, as body weight of a fish is not always proportional to the cube of its total length [63]. The findings of the present study are in general agreement with those reported with Naeem and Ishtiaq [60] in wild *Mystus bleekeri*, Khalid and Naeem [64] in farmed *Ctenophyrngodon idella* and Kousar et al. [45] in Genetically Improved Farmed Tilapia.

As, higher cost of a fish feed containing higher crude protein is considered a limitations or challenge in implementing the recommended 40% crude protein diet on a commercial scale, but farming industry should not compromise as it influences positively on the growth performance, FCR and quality of fish as food. Moreover, delving the further research is recommended to explore the lasting effects of employing the optimal 40% crude protein diet on *C. batrachus*, considering aspects related to reproduction and overall health.

## 5. Conclusion

This investigation specifies that dietary protein levels affect the growth, FCR and proximate composition of *Clarias batrachus*. The best growth parameters and chemical composition (containing highest level of protein, mineral and fat constituents) can be achieved by feeding the catfish with a diet comprising 40% crude protein (CP) than other dietary protein levels (25%, 30%, 35%, 45% or 50% CP), and consequently, it is suggested that addition of 40% protein in feed is ideal for growth and effective feed utilization of the walking catfish, *C. batrachus*. It is also evident that despite the variations, the mean values of percentage protein in different treatments of this study indicates that *C. batrachus* is a good source of protein to consumers. Moreover, body size shows a pronounced impact on body composition of fish. Data produced in the current research would be beneficial in evolving nutritionally balanced diets for the semi-intensive and intensive culture of this catfish.

## Author Contributions

**Conceptualization:** Amina Zuberi, Muhammad Ali.

**Data curation:** Zara Naeem, Amina Zuberi, Ammar Danyal Naeem, Muhammad Naeem.

**Formal analysis:** Zara Naeem, Muhammad Naeem.

**Investigation:** Zara Naeem.

**Methodology:** Zara Naeem, Ammar Danyal Naeem, Muhammad Naeem.

**Project administration:** Amina Zuberi.

**Resources:** Ammar Danyal Naeem.

**Software:** Zara Naeem.

**Supervision:** Amina Zuberi.

**Validation:** Zara Naeem.

**Visualization:** Zara Naeem, Muhammad Ali.

**Writing – original draft:** Zara Naeem.

**Writing – review & editing:** Zara Naeem, Amina Zuberi, Muhammad Ali, Muhammad Naeem.

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
