## [Decision Letter · Decision Letter 0]

9 Feb 2024

PONE-D-23-43259

An approach to optimizing dietary protein to growth and body composition in walking catfish, Clarias batrachus (Linneaeus, 1758)

PLOS ONE

Dear Dr. Naeem,

Thank you for submitting your manuscript to PLOS ONE. After careful consideration, we feel that it has merit but does not fully meet PLOS ONE’s publication criteria as it currently stands. Therefore, we invite you to submit a revised version of the manuscript that addresses the points raised during the review process.

We look forward to receiving your revised manuscript.

Kind regards,

Amit Ranjan, Ph.D.

Academic Editor

PLOS ONE

Journal Requirements:

Reviewers' comments:

Reviewer's Responses to Questions

**Comments to the Author**

1. Is the manuscript technically sound, and do the data support the conclusions?

Reviewer #1: Partly

Reviewer #2: Partly

Reviewer #3: Yes

Reviewer #4: Yes

2. Has the statistical analysis been performed appropriately and rigorously? 

Reviewer #1: Yes

Reviewer #2: I Don't Know

Reviewer #3: Yes

Reviewer #4: Yes

3. Have the authors made all data underlying the findings in their manuscript fully available?

Reviewer #1: Yes

Reviewer #2: No

Reviewer #3: Yes

Reviewer #4: Yes

4. Is the manuscript presented in an intelligible fashion and written in standard English?

Reviewer #1: Yes

Reviewer #2: No

Reviewer #3: No

Reviewer #4: Yes

5. Review Comments to the Author

Reviewer #1: The manuscript entitled “An approach to optimizing dietary protein to growth and body composition in walking catfish, Clarias batrachus (Linneaeus, 1758)” explores the best or most effective levels of dietary protein to enhance the growth and influence the body composition of walking catfish. This manuscript lacking the necessary details like dietary ingredients used for diet preparation, protein source etc. I feel this paper is not suitable to be published on the PLOS ONE in the current state. I have noted some major comments as follows:

1. I would like to emphasize the importance of explicitly disclosing the dietary ingredients employed in the diet preparations outlined in this study. Additionally, it is crucial for the authors to clearly specify the protein source utilized in the study. This information is essential for ensuring transparency, reproducibility, and a comprehensive understanding of the experimental conditions. Please provide a detailed account of the dietary components and protein source to strengthen the integrity of the research

2. I suggest that the authors also include the proximate compositions of the experimental diets. Providing this information would add valuable detail to the study, aiding in the understanding of nutritional aspects and ensuring transparency for readers.

3. It would be beneficial to provide the instrument names, manufacturers, and any relevant specifications in the Materials and Methods section.

4. In line 113-114, the authors mention drying temp at 80ºC for water content analysis. However, it is essential to clarify the duration of this drying process.

5. I suggest the authors reconsider the use of the difference method for calculating protein content in the manuscript. While this method provides an approximate estimate, for more accurate and reliable results, it is advisable to employ established laboratory analysis method such as the Kjeldahl method. These methods are widely accepted for protein determination and would enhance the precision and credibility of the protein content reported in the study.

6. I suggest that the authors consider investigating protein digestibility as part of this study. Including an analysis of protein digestibility would provide valuable insights into the bioavailability of dietary proteins and contribute to a more comprehensive understanding of the nutritional aspects under investigation.

7. On page 9, line 287, there appears to be a typo in the scientific name. It should be 'Ctenopharyngodon idella' instead of 'Ctenophyrngodon Idella.' Please correct this to ensure accuracy in the scientific nomenclature throughout the manuscript.

8. There few typesetting errors throughout the manuscript that should be corrected accordingly

Reviewer #2: The main purpose of the article appears to be to investigate the effects of varying levels of dietary protein on the survival, growth parameters, and proximate composition of Clarias batrachus, a commercially important food fish in Pakistan. While the study offers some insights, there are notable shortcomings that raise questions about its suitability for publication in its current form. My reasons are as follows:

1. Insufficient novelty and significance. The researchers evaluate different crude protein percentages in the diets and assess their impact on factors such as weight gain, daily growth rate, protein efficiency ratio, and body composition, which are merely the most fundamental parameters used to evaluate feed formulations or to confirm the appropriate proportions of certain ingredients, and the work has already been done in closely related species. Since the experimental limitations and the lack of further investigation, a thorough and deep discussion could not be provided in the manuscript. Even the current data were not properly analyzed and discussed. It only indicated whether the results aligned or diverged from previous studies, without providing an explanation for the observed outcomes. To foster a more comprehensive discussion on this topic, additional research should be undertaken to explore factors such as enzyme activity, oxidative stress, gut microbiota, and blood physiological indicators, even immune response level.

2. The introduction was not well written. No former studies about the function of crude proteins (or other nutritional ingredients) in upgrading biological traits in cultivated fish were introduced, which had already been widely studied and was supposed to be the most important information in this part.

3. The authors should also provide more detailed information on the statistical methods used, as well as any potential limitations and sources of bias in their study.

4. Need thorough language editing.

Reviewer #3: The present manuscript reported a well-defined experimental design that documented using simple and solid data (statistically confirmed), which is the best quantity of dietary protein in walking catfish farming.

The work is generally written in an understandable way, however there are several parts that are more confusing and deserve a grammatical and syntactic revision. I report here a list of lines that should be reformulated to better clarify the concepts expressed:

Lines 40-42; 60-61; 71-73; 200-206; 258-264.

Important materials that must be provided in this manuscript concern: a complete table regarding the composition of the diets, with important emphasis on the origin (animal/vegetable) of the protein source used, and its amino acid composition.

Along the text I also encountered several errors and inaccuracies. I reported them below:

Line 14: "Clarias batrachus is commercialy..." should be corrected as "Clarias batrachus is a commercially..."

Line 17: the symbol @ should be corrected with the symbol ≈ or ±.

Line 31: "insignificant correlation" should be changed to "non-significant correlation".

Line 47: the term "realized" is not appropriate, I suggest to use "considered" or "represent".

Line 53: "A vital issue" is not properly appropriate in this context, I suggest to change it.

Line 68: "Henceforth" should be changed to "Therefore".

Lines 75-78: I suggest to reformualte the sentence in this way: "Fish proximate body composition constituents (fat, protein, water, organic content and ash) are influenced by diet, feed rate, sex, genetic strain, age, species and also by changing body size and condition factor [26-28].

Line 93: the same correction reported for line 17.

In material and methods should be also provided information regarding the illumination (light/dark cycle) during the trial.

Line 129: the same correction reported for line 31.

Line 133: "Significant highest" should be changed to "Significantly highest".

Lines 150-152: I suggest to reformualte the sentence in this way: "Mean value of water, ash, fat and protein contents (% wet mass) were ranged from 74.10±0.31% to 79.23±0.52%, 3.12±0.07% - 4.68±0.05%, 3.90±0.06% - 4.43±0.05% and 13.09±0.58% - 16.79±0.27% in the studied treatments..." to avoid repetitions.

Line 155: "and T6 than T4" should be changed to "and T6 than for T4".

Lines 201-202: I suggest to reformualte the sentence in this way: "the present experiment was found similar to that previously described by Tadesse [30]" to avoid repetitions.

Line 207: "is a vital gauge".

Line 210: Although FCR value between 1.2 and 1.5 represent a good indicators, it could not be used as absolute standard in all fish species, as it can change according to several culturing factors. In addition the reference [33] reported this range in tilapia. I suggest to reformulate this sentence.

Line 216: "animal ingredients" it should be clarified in M&M, providing further information about this source.

Line 281: there is a repetition of "in".

REFERENCES should be revised and checked accordingly to the journal requirements as I encountered some errors, such as Line 392, 400 and 416.

Table 1 and 2: It is not an error, however I suggest to substitute the title of the last column from "Sig." to "p-value", and the ".000" with "< .001".

In Table 2 the caption is missing.

Reviewer #4: An approach to optimizing dietary protein to growth and body composition in walking catfish, Clarias batrachus (Linneaeus, 1758)

This study (PONE-D-23-43259) examines the repercussions of varying protein levels in the diet of Clarias batrachus, a commercially significant food fish in Pakistan. The research assesses the influence of different protein levels on the survival, growth parameters, and proximate composition of C. batrachus throughout a 90-day period. While acknowledging the quality of your paper, I have some suggestions to enhance your work.

- I recommend delving into further research to explore the lasting effects of employing the optimal 40% crude protein diet on C. batrachus, considering aspects related to reproduction and overall health.

- Additionally, it is advisable to investigate whether environmental factors, such as water quality or temperature, played a contributory role in the observed outcomes, and to evaluate the necessity of incorporating these factors into future studies.

- Furthermore, I propose that the author undertake a comparative analysis of the findings from this study with analogous research on other fish species to enrich the discussion section.

- Moreover, conducting a comprehensive economic analysis would contribute to evaluating the cost-effectiveness of implementing the 40% crude protein diet in aquaculture in comparison to alternative protein levels.

- Finally, I suggest reflecting on strategies to effectively disseminate the discoveries of this study to aquaculture practitioners, farmers, and stakeholders. Exploring avenues for promoting optimal dietary practices within these target audiences could be beneficial.

I ask the authors to edit and respond to these points:

1. Some sentences were not clear, you must formulate them correctly. Yet, English should be improved and reviewed by a native speaker.

2. You should be precise about what factors might contribute to the 100% survival rate across all treatments.

3. Could the study provide insights into why treatment with 40% crude protein (T4) resulted in the highest growth parameters for the studied specie?

4. What implications do the variations in water, ash, fat, and protein content have for the nutritional value of C. batrachus in different treatments?

5. How might the observed differences in body composition across treatments impact the quality of the fish as a food source?

6. Why did the treatment with 40% crude protein (T4) exhibit the best Feed Conversion Ratio (FCR)? Are there specific physiological or metabolic reasons for this result?

7. How does body size affect the body composition of C. batrachus?

8. In what ways does the condition factor correlate with body size, and why were these correlations mostly insignificant?

9. How can the findings of this study be practically applied in aquaculture operations, especially in formulating optimal diets for C. batrachus?

10. Are there any limitations or challenges in implementing the recommended 40% crude protein diet on a commercial scale?

6. PLOS authors have the option to publish the peer review history of their article (what does this mean?). If published, this will include your full peer review and any attached files.

Reviewer #1: **Yes: **Nitesh Kumar Yadav

Reviewer #2: No

Reviewer #3: **Yes: **Federico Moroni

Reviewer #4: **Yes: **Sami MILI

---

## [Author Response · Author response to Decision Letter 0]

17 Feb 2024

Dear Sir,

The manuscript entitled “An approach to optimizing dietary protein to growth and body composition in walking catfish, Clarias batrachus (Linneaeus, 1758)” is improved/revised following the valuable suggestions of the reviewers.

All the corrections/changes are incorporated in the manuscript, and detailed point-to-point response is mentioned in the following table. 

Response to Editor

1. Ethics statement should only appear in the Methods section of your manuscript. If your ethics statement is written in any section besides the Methods, please move it to the Methods section and delete it from any other section. Please ensure that your ethics statement is included in your manuscript, as the ethics statement entered into the online submission form will not be published alongside your manuscript.

Author Response:

Added

2. In the online submission form, you indicated that

"The datasets used and analysed during the current study are available from the corresponding author on reasonable request.".

3. Uploaded as supplementary information.

Author Response:

All data underlying the findings described in their manuscript to be freely available to other researchers after acceptance of paper, either

3. Uploaded as supplementary information

3. Please amend your list of authors on the manuscript to ensure that each author is linked to an affiliation.

We note that you have included affiliation numbers 1 and 2 however only affiliation1 have authors linked to them. Please amend affiliation 2 to link an author to it or remove if added in error.

Author Response:

Author list amended.

4. 4. We are uncertain of our previous requests. To comply with PLOS ONE submissions requirements, in your Methods section, please provide additional information regarding the experiments involving animals and ensure you have included details on (1) methods of sacrifice, (2) methods of anesthesia and/or analgesia, and (3) efforts to alleviate suffering.

Author Response:

Added

REVIEWER 1

1. I would like to emphasize the importance of explicitly disclosing the dietary ingredients employed in the diet preparations outlined in this study. Additionally, it is crucial for the authors to clearly specify the protein source utilized in the study. This information is essential for ensuring transparency, reproducibility, and a comprehensive understanding of the experimental conditions. Please provide a detailed account of the dietary components and protein source to strengthen the integrity of the research. 

Author Response:

Detailed dietary components of experimental feeds added as a new table (Table 1).

2. I suggest that the authors also include the proximate compositions of the experimental diets. Providing this information would add valuable detail to the study, aiding in the understanding of nutritional aspects and ensuring transparency for readers. 

Author Response:

Proximate composition of the experimental diets is also added in Table 1.

3. It would be beneficial to provide the instrument names, manufacturers, and any relevant specifications in the Materials and Methods section. 

Author Response

Instrument names and manufacturersare added in the Materials and Methods section.

4. In line 113-114, the authors mention drying temp at 80ºC for water content analysis. However, it is essential to clarify the duration of this drying process. 

Author Response

Mentioned in the manuscript. 

“Till constant of body weight”

5. I suggest the authors reconsider the use of the difference method for calculating protein content in the manuscript. While this method provides an approximate estimate, for more accurate and reliable results, it is advisable to employ established laboratory analysis method such as the Kjeldahl method. These methods are widely accepted for protein determination and would enhance the precision and credibility of the protein content reported in the study. 

Author Response

Difference method was used to evaluate the protein contents in this study. However, valuable suggestion of the reviewer will be considered for future studies.

6. I suggest that the authors consider investigating protein digestibility as part of this study. Including an analysis of protein digestibility would provide valuable insights into the bioavailability of dietary proteins and contribute to a more comprehensive understanding of the nutritional aspects under investigation. 

Author Response:

Protein digestibility was not the part of this study. However, valuable suggestion of the reviewer will be considered for future studies.

7. On page 9, line 287, there appears to be a typo in the scientific name. It should be 'Ctenopharyngodon idella' instead of 'Ctenophyrngodon Idella.' Please correct this to ensure accuracy in the scientific nomenclature throughout the manuscript. 

Author Response

Scientific name is corrected.

8. There few typesetting errors throughout the manuscript that should be corrected accordingly. 

Author Response

Typesetting errors throughout the manuscript are corrected.

Reviewer 2

1. Insufficient novelty and significance. The researchers evaluate different crude protein percentages in the diets and assess their impact on factors such as weight gain, daily growth rate, protein efficiency ratio, and body composition, which are merely the most fundamental parameters used to evaluate feed formulations or to confirm the appropriate proportions of certain ingredients, and the work has already been done in closely related species. Since the experimental limitations and the lack of further investigation, a thorough and deep discussion could not be provided in the manuscript. Even the current data were not properly analyzed and discussed. It only indicated whether the results aligned or diverged from previous studies, without providing an explanation for the observed outcomes. To foster a more comprehensive discussion on this topic, additional research should be undertaken to explore factors such as enzyme activity, oxidative stress, gut microbiota, and blood physiological indicators, even immune response level. 

Author Response

Discussion is improved highlighting the novelty and significance of the study.

Explanation of observed outcomes of the study is added.

2. The introduction was not well written. No former studies about the function of crude proteins (or other nutritional ingredients) in upgrading biological traits in cultivated fish were introduced, which had already been widely studied and was supposed to be the most important information in this part. 

Author Response

Introduction is improved.

3. The authors should also provide more detailed information on the statistical methods used, as well as any potential limitations and sources of bias in their study. 

Author Response

Detail is added

4. Need thorough language editing. 

Author Response

Language is improved.

Reviewer 3

1. The work is generally written in an understandable way, however there are several parts that are more confusing and deserve a grammatical and syntactic revision.

 I report here a list of lines that should be reformulated to better clarify the concepts expressed:

Lines 40-42; 60-61; 71-73; 200-206; 258-264. 

Author Response

Lines 40-42; 60-61; 71-73; 200-206; 258-264 are reformulated to make the statements more clear.

2. Important materials that must be provided in this manuscript concern: a complete table regarding the composition of the diets, with important emphasis on the origin (animal/vegetable) of the protein source used, and its amino acid composition. 

3. Line 14: "Clarias batrachus is commercialy..." should be corrected as "Clarias batrachus is a commercially..." 

Author Response

Corrected as suggested by the reviewer.

4. Line 17: the symbol @ should be corrected with the symbol ≈ or ±. 

Author Response

@ is replaced with “at the rate of”

5. Line 31: "insignificant correlation" should be changed to "non-significant correlation". 

Author Response

“insignificant correlation" is changed to "non-significant correlation”

6. Line 47: the term "realized" is not appropriate, I suggest to use "considered" or "represent". 

Author Response

“realized” is replaced with "considered"

7. Line 53: "A vital issue" is not properly appropriate in this context, I suggest to change it. 

Author Response

"A vital issue" is replaced with “An important issue”

8. Line 68: "Henceforth" should be changed to "Therefore". 

Author Response

"Henceforth" is changed to "Therefore".

9. Lines 75-78: I suggest to reformualte the sentence in this way: "Fish proximate body composition constituents (fat, protein, water, organic content and ash) are influenced by diet, feed rate, sex, genetic strain, age, species and also by changing body size and condition factor [26-28]. 

Author Response

Corrected as suggested by the reviewer.

10. Line 93: the same correction reported for line 17. 

Author Response

@ is replaced with “at the rate of”

11. In material and methods should be also provided information regarding the illumination (light/dark cycle) during the trial. 

Author Response

Light/dark cycle during the trial is added in MM section.

12. Line 129: the same correction reported for line 31. 

Author Response

“insignificant correlation" is changed to "non-significant correlation”

13. Line 133: "Significant highest" should be changed to "Significantly highest". 

Author Response

"Significant highest" is changed to "Significantly highest".

14. Lines 150-152: I suggest to reformualte the sentence in this way: "Mean value of water, ash, fat and protein contents (% wet mass) were ranged from 74.10±0.31% to 79.23±0.52%, 3.12±0.07% - 4.68±0.05%, 3.90±0.06% - 4.43±0.05% and 13.09±0.58% - 16.79±0.27% in the studied treatments..." to avoid repetitions. 

Author Response

Reformulated as suggested by the reviewer.

15. Line 155: "and T6 than T4" should be changed to "and T6 than for T4". 

Author Response

"and T6 than T4" is changed to "and T6 than for T4".

16. Lines 201-202: I suggest to reformualte the sentence in this way: "the present experiment was found similar to that previously described by Tadesse [30]" to avoid repetitions. 

Author Response

Reformulated as suggested by the reviewer.

17. Line 207: "is a vital gauge". 

Author Response

Corrected by adding space.

Line 210: Although FCR value between 1.2 and 1.5 represent a good indicators, it could not be used as absolute standard in all fish species, as it can change according to several culturing factors. In addition the reference [33] reported this range in tilapia. I suggest to reformulate this sentence. 

Author Response

Reformulated.

Moreover, the following lines are added in Discussion for make it more clear:

Tadesse [30] reported the lowest feed conversion ratio (FCR=2.1) in fish fed with 40% CP, indicating its suitability for African catfish fingerlings than the other test diets. 

18. Line 216: "animal ingredients" it should be clarified in M&M, providing further information about this source. 

Author Response

The line is reformulated to make clear. 

Moreover, detailed dietary components of experimental feeds added as Table 1.

19. Line 281: there is a repetition of "in". 

Author Response 

Corrected.

REFERENCES should be revised and checked accordingly to the journal requirements as I encountered some errors, such as Line 392, 400 and 416. 

Author Response

Corrected.

20. Table 1 and 2: It is not an error, however I suggest to substitute the title of the last column from "Sig." to "p-value", and the ".000" with "< .001". In Table 2 the caption is missing. 

Author Response

Table 1: "Sig." is substituted to "p-value", and the ".000" with "< .001".

Table 2: Footnote is added.

Reviewer 4

1. Some sentences were not clear, you must formulate them correctly. Yet, English should be improved and reviewed by a native speaker. 

Author Response

Language of the manuscript is improved. 

2. You should be precise about what factors might contribute to the 100% survival rate across all treatments. 

Author Response

Mentioned in the discussion section.

“…survival rate was recorded as 100% in all of the studied groups, indicating high tolerance of the fish in the confined system and also represents that rearing conditions were good (optimal).”

3. Could the study provide insights into why treatment with 40% crude protein (T4) resulted in the highest growth parameters for the studied specie? 

Author Response

It may be due to the high proportion of protein derived from easily digestible animal ingredients, as mentioned in discussion.

4. What implications do the variations in water, ash, fat, and protein content have for the nutritional value of C. batrachus in different treatments? 

Author Response

Results showed that feeding appropriate diet (containing 40% CP) to the fish resulted higher protein and lipid content in fish. 

Thus following statement is added in the discussion:

Hence, the results indicated that farmer can achieve not only higher growth and low FCR but can attain higher protein and lipid contents in the C. batrachus by feeding the fish 40% crude protein, rather higher crude protein (45%CP) in diet.

5. How might the observed differences in body composition across treatments impact the quality of the fish as a food source? 

Author Response

As a higher proportion of protein content may exhibits the good quality of the fish as a food source.

6. Why did the treatment with 40% crude protein (T4) exhibit the best Feed Conversion Ratio (FCR)? Are there specific physiological or metabolic reasons for this result? 

Author Response

It may be due to the high proportion of protein derived from easily digestible animal ingredients, as mentioned in discussion.

7. How does body size affect the body composition of C. batrachus? 

Author Response

Described in the discussion section.

“Literature shows that body weight of a fish influences the various body constituents [16], and fat and protein increase while ash and water decrease with the increasing total length [56].”

8. In what ways does the condition factor correlate with body size, and why were these correlations mostly insignificant? 

Author Response

Mentioned in the discussion section.

“…condition factor, as body weight of a fish is not always proportional to the cube of its total length [60]”

9. How can the findings of this study be practically applied in aquaculture operations, especially in formulating optimal diets for C. batrachus? 

Author Response

Mentioned in the manuscript.

“Farmer can achieve not only higher growth and low FCR but can attain higher protein and lipid contents in the C. batrachus by feeding the fish 40% crude protein, rather higher crude protein (45%CP) in diet.”

“Data produced in the current research would be beneficial in evolving nutritionally balanced diets for the semi-intensive and intensive culture of this catfish.”

10. Are there any limitations or challenges in implementing the recommended 40% crude protein diet on a commercial scale? 

Author Response

Added in the discussion:

As, higher cost of a fish feed containing higher crude protein is considered a limitations or challenge in implementing the recommended 40% crude protein diet on a commercial scale, but farming industry should not compromise as it influences positively on the growth performance, FCR and quality of fish as food.

---

## [Decision Letter · Decision Letter 1]

11 Mar 2024

PONE-D-23-43259R1An approach to optimizing dietary protein to growth and body composition in walking catfish, Clarias batrachus (Linneaeus, 1758)PLOS ONE

Dear Dr. Naeem,

Thank you for submitting your manuscript to PLOS ONE. After careful consideration, we feel that it has merit but does not fully meet PLOS ONE’s publication criteria as it currently stands. Therefore, we invite you to submit a revised version of the manuscript that addresses the points raised during the review process.

We look forward to receiving your revised manuscript.

Kind regards,

Amit Ranjan, Ph.D.

Academic Editor

PLOS ONE

Journal Requirements:

Reviewers' comments:

Reviewer's Responses to Questions

**Comments to the Author**

1. If the authors have adequately addressed your comments raised in a previous round of review and you feel that this manuscript is now acceptable for publication, you may indicate that here to bypass the “Comments to the Author” section, enter your conflict of interest statement in the “Confidential to Editor” section, and submit your "Accept" recommendation.

Reviewer #3: (No Response)

Reviewer #4: All comments have been addressed

2. Is the manuscript technically sound, and do the data support the conclusions?

Reviewer #3: Yes

Reviewer #4: Yes

3. Has the statistical analysis been performed appropriately and rigorously? 

Reviewer #3: Yes

Reviewer #4: Yes

4. Have the authors made all data underlying the findings in their manuscript fully available?

Reviewer #3: Yes

Reviewer #4: Yes

5. Is the manuscript presented in an intelligible fashion and written in standard English?

Reviewer #3: Yes

Reviewer #4: Yes

6. Review Comments to the Author

Reviewer #3: The authors resolved several errors and missing information during the firast round of review. However, there are few things that should be adressed.

Line 64-65: The correction made did not include "may BE higher". In addition, this part seems a bit confusiong, the author should clarify the concept better. FCR is a commonly used measure of conversion which is commonly used over the entire production life of a fish, and obviously, the amount of feed changes during this period, but this does not invalidate its importance.

Line 107: The two new references 29 and 30 have incorrect numbers referring with the list of references, please check them. In addition, although accurate, the reference Preston et al.,2016, used for the calculation of proximate composition, does not include all the information for the correct evaluation of the ingredients listed in Table 1. Please provide further details or change the reference.

Line 218: "feeding fish WITH 30%..."

Table 1. There is no indication about the unity of measurement of the ingredients. It must be indicated, together with the acronym CMC, in the caption.

Furthermore, regarding the reported CP values, there are inconsistencies between the values and the name of the experimental diets for CP-35; CP-40; CP-45 and CP-50 (for these diets the effective amount of crude protein is lower). Since this part represents the core of the article and the author in the discussion and conclusion sections refers to the diet as the effective amount of crude protein, this information needs to be clarified and corrected.

Reviewer #4: (No Response)

7. PLOS authors have the option to publish the peer review history of their article (what does this mean?). If published, this will include your full peer review and any attached files.

Reviewer #3: **Yes: **Federico Moroni

Reviewer #4: **Yes: **Sami MILI

---

## [Author Response · Author response to Decision Letter 1]

13 Mar 2024

Dear Sir,

The manuscript entitled “An approach to optimizing dietary protein to growth and body composition in walking catfish, Clarias batrachus (Linneaeus, 1758)” is improved/revised following the valuable suggestions of the reviewer.

All the corrections/changes are incorporated in the manuscript, and detailed point-to-point response is mentioned in the following table. 

Reviewer Comment

1. Line 64-65: The correction made did not include "may BE higher". In addition, this part seems a bit confusiong, the author should clarify the concept better. FCR is a commonly used measure of conversion which is commonly used over the entire production life of a fish, and obviously, the amount of feed changes during this period, but this does not invalidate its importance.

Author Response

Clarified Line 64-65. 

2. Line 107: The two new references 29 and 30 have incorrect numbers referring with the list of references, please check them. In addition, although accurate, the reference Preston et al.,2016, used for the calculation of proximate composition, does not include all the information for the correct evaluation of the ingredients listed in Table 1. Please provide further details or change the reference.

Author Response

Numbering of references 29 and 30 are corrected.

Another reference is added for the calculation of proximate composition in Table 1.

3. Line 218: "feeding fish WITH 30%..."

Author Response

Corrected Line 218

4. Table 1. There is no indication about the unity of measurement of the ingredients. It must be indicated, together with the acronym CMC, in the caption.

Furthermore, regarding the reported CP values, there are inconsistencies between the values and the name of the experimental diets for CP-35; CP-40; CP-45 and CP-50 (for these diets the effective amount of crude protein is lower). Since this part represents the core of the article and the author in the discussion and conclusion sections refers to the diet as the effective amount of crude protein, this information needs to be clarified and corrected.

Author Response

Unit (%) added in the Table title.

Full form of CMC is added.

Format of treatment names made consistent and changed in discussion and conclusion section accordingly.

---

## [Editor Report · Decision Letter 2]

20 Mar 2024

An approach to optimizing dietary protein to growth and body composition in walking catfish, Clarias batrachus (Linneaeus, 1758)

PONE-D-23-43259R2

Dear Dr. Naeem,

We’re pleased to inform you that your manuscript has been judged scientifically suitable for publication and will be formally accepted for publication once it meets all outstanding technical requirements.

Kind regards,

Amit Ranjan, Ph.D.

Academic Editor

PLOS ONE